# Discovery of Late Permian Adakite in Eastern Central Asian Orogenic Belt: Implications for Tectonic Evolution of Paleo-Asian Ocean

Haihua Zhang [1,2], Liang Qiu [3,*], Jian Zhang [1], Yongfei Ma [4], Yujin Zhang [1], Shuwang Chen [1], Huiliang Dong [5] and Yuejuan Zheng [2]

1   Shenyang Center of China Geological Survey, Shenyang 110034, China; zhanghaihua311@mail.ustc.edu.cn (H.Z.); zhangjian@mail.cgs.gov.cn (J.Z.); syzhangyujin@163.com (Y.Z.); sycswgeology@163.com (S.C.)
2   CAS Key Laboratory of Crust-Mantle Materials and Environments, School of Earth and Space Sciences, University of Science and Technology of China, Hefei 230026, China; zhengyueliang666@163.com
3   State Key Laboratory of Geological Processes and Mineral Resources, School of Earth Sciences and Resources, China University of Geosciences, Beijing 100083, China
4   College of Earth Sciences, Institute of Disaster Prevention, Sanhe 065201, China; mayongfei@cidp.edu.cn
5   Gaocheng No. 3 Middle School, Shijiazhuang 052160, China; huaxuelang@163.com
*   Correspondence: qiul@cugb.edu.cn

**Abstract:** The Permian to Triassic period represents a pivotal phase in the evolution of the Paleo-Asian Ocean, marked by significant tectonic transitions from subduction, collision, and post-orogenic extension. The timing of closure of the Paleo-Asian Ocean in northeastern China has always been controversial. In this contribution, the petrology, zircon U-Pb geochronology and geochemistry are conducted on granite found in well HFD1, Songliao Basin, eastern part of Central Asian orogenic belt. Zircon U-Pb dating indicates that granite crystallized at $258.9 \pm 2.2$ Ma, as the product of magmatism occurred in the early Late Permian. The rocks have high $SiO_2$, $Al_2O_3$, $Na_2O$ content, negative Eu anomaly, light enrichment of rare-earth elements, depletion of heavy rare-earth elements, high Sr (448.29–533.11 ppm, average 499.68 ppm), low Yb (0.49–0.59 ppm, average 0.54 ppm), Y (4.23–5.19 ppm, average 4.49 ppm), and high Sr/Y ratios (98–125, average 112) and can be classified as O-type adakite. This is the first discovery of late Paleozoic adakite in the Songliao Basin and the neighboring areas. The geochemistry of adakite indicates derivation by partial melting of MORB-type subducted oceanic crust, indicating that the subduction of the Paleo-Asian Oceanic lithosphere lasted until at least 258.9 Ma.

**Keywords:** Late Permian; adakite; Paleo-Asian Ocean; geochemistry; tectonic setting





## 1. Introduction

The Central Asian orogenic belt (CAOB), situated between the Siberia, Europe and North China-Tarim blocks, is the largest accretionary orogenic belt in the world. Extending from the Ural Mountains to the western Pacific coast, the CAOB traverses regions including Kazakhstan, the Tianshan Mountains, the Altai Mountains, Mongolia, and northeastern China [1–6]. This belt, formed through multiple episodes of accretion within the Paleo-Asian Ocean, comprises a complex structure of magmatic arcs, accretionary terranes, arc-related basins, and microcontinents [3–5,7–9].

The eastern segment of the CAOB contains multiple micro-blocks with Precambrian ancient basement, including the Erguna, the Xing'an, the Xilinhot-Songliao, and the Jia-musi micro-blocks, which amalgamated before the latest Paleozoic. These micro-blocks collided with the North China Block at the end of the late Paleozoic, resulting in the closure of the Paleo-Asian Ocean and initiating a post-collisional extensional orogenic stage in northeastern China (Figure 1) [6,10–12].

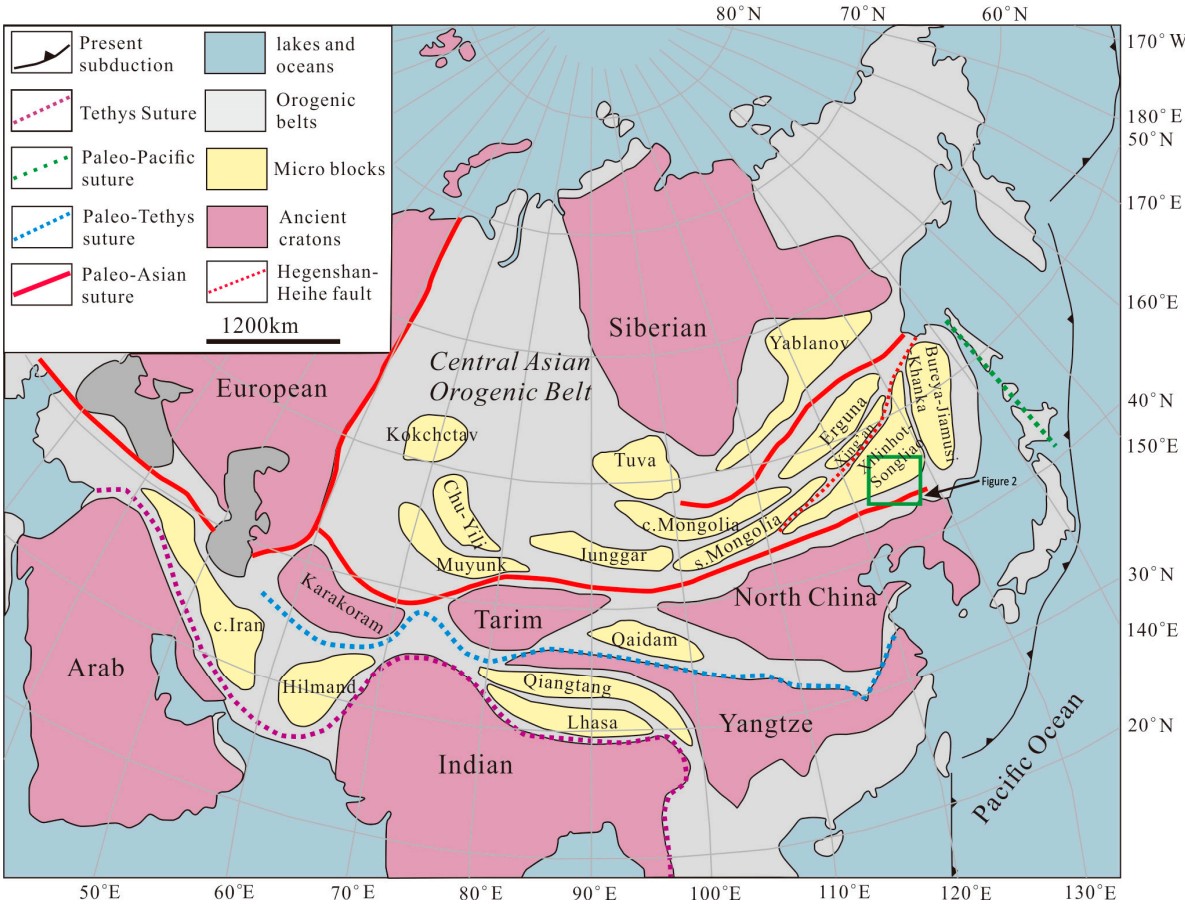

**Figure 1.** Tectonic sketch of Eurasia continent (modified from [13]).

The exact location and timing of suturing between the Siberia and North China Blocks have been a subject of debate. Most scholars consider the Solonker-Xar Moron-Changchun-Yanji line as the suture zone [14–18], while others argue for the He'genshan-Heihe fault zone [19–22]. There are also differing views on the final closure time of the Paleo-Asian Ocean, ranging from the Middle Devonian [20,23,24] to the Late Devonian to Early Carboniferous [25–27], with most scholars leaning towards the Middle-Late Permian to Early-Middle Triassic [11,15,28–32].

Late Permian is a crucial period for the transition from the Paleo-Asian Oceanic tectonic domain to the Circum-Pacific tectonic domain [29,33]. The Songliao Basin is a large oil and gas basin in China, but research on its pre-Mesozoic tectonic evolution and deep magma events is scarce, particularly regarding Late Paleozoic magma events [34–36]. Gao et al. [35] studied granites in the southern part of the Songliao Basin, suggesting that Middle Jurassic magmatic activity formed the majority of the basement granites. Yu et al. [36] studied volcanic rocks in the northern part of the Songliao Basin, confirming multiple magmatic thermal events from the Paleozoic to the Mesozoic, with the Jurassic and Triassic volcanic activities being the most frequent, which might be related to the collision of Northeast blocks and the collision and suturing of the North China Block and Siberia Block at the end of the Permian. However, due to the limited exposure of bedrock in the Songliao Basin and the challenges in obtaining samples and data from deep within the basin, previous research on Late Paleozoic magmatic and tectonic evolution has focused mainly on the peripheral outcrop areas. Studies on the Songliao Basin, particularly isotopic and geochemical studies, are lacking. Therefore, this study aims to investigate intrusive rocks in the HFD1 core from the Songliao Basin to determine their precise ages and discuss their petrogenesis and dynamic background, to understand the connection between the magmatism and evolution of the Paleo-Asian Ocean in the Songliao Basin.

## 2. Geological Setting

Northeastern China, situated in the eastern Central Asian Orogenic Belt, underwent a complex tectonic evolution history, which was influenced by the overlay of the Paleozoic Paleo-Asian Oceanic tectonic domain, the Mesozoic Mongol-Okhotsk Oceanic tectonic domain, and the Cenozoic Circum-Pacific tectonic domain [2,6,12,13,37,38].

This study area is located in the eastern CAOB, in the Songliao Basin of China, belonging to the Songliao-Xilinhot block, which is a structurally complex region sandwiched between the Siberian Plate, the North China Block, and the Pacific Plate, and is a key area for understanding the tectonic evolution of the Siberia Block, North China Block, and the Paleo-Asian Ocean. The basement comprises Paleozoic Carboniferous-Permian sedimentary cover and various phases of felsic magma emplacement (Figure 2a). During the core logging process of the HFD1 well in the study area, we discovered intrusions of granite, which represent the subject of this study (Figure 2b). The well reveals the stratigraphy from bottom to top: Permian Linxi Formation ($P_3l$) from 2089.4 to 1386.4 m, and the lithology is mainly siltstone and mudstone; Cretaceous Quantou Formation ($K_1q$) from 1386.4 to 1181.8 m, and the lithology is mainly composed of conglomerate and sandstone; Qingshankou Formation ($K_2qn$) from 1181.8 to 1083.0 m and mainly composed of mudstone; Yaojia Formation ($K_2y$) from 1083.0 to 876.0 m, and the lithology is mainly siltstone; Nenjiang Formation ($K_2n$) from 876.0 to 529.40 m and mainly composed of mudstone; Sifangtai Formation ($K_2s$) from 529.4 to 473.1 m, and the lithology is medium-fine sandstone; Mingshui Formation ($K_2m$) from 473.1 to 202.4 m and mainly composed of sandstone; and Quaternary from 202.4 to 0 m and composed mainly of loose sediments.

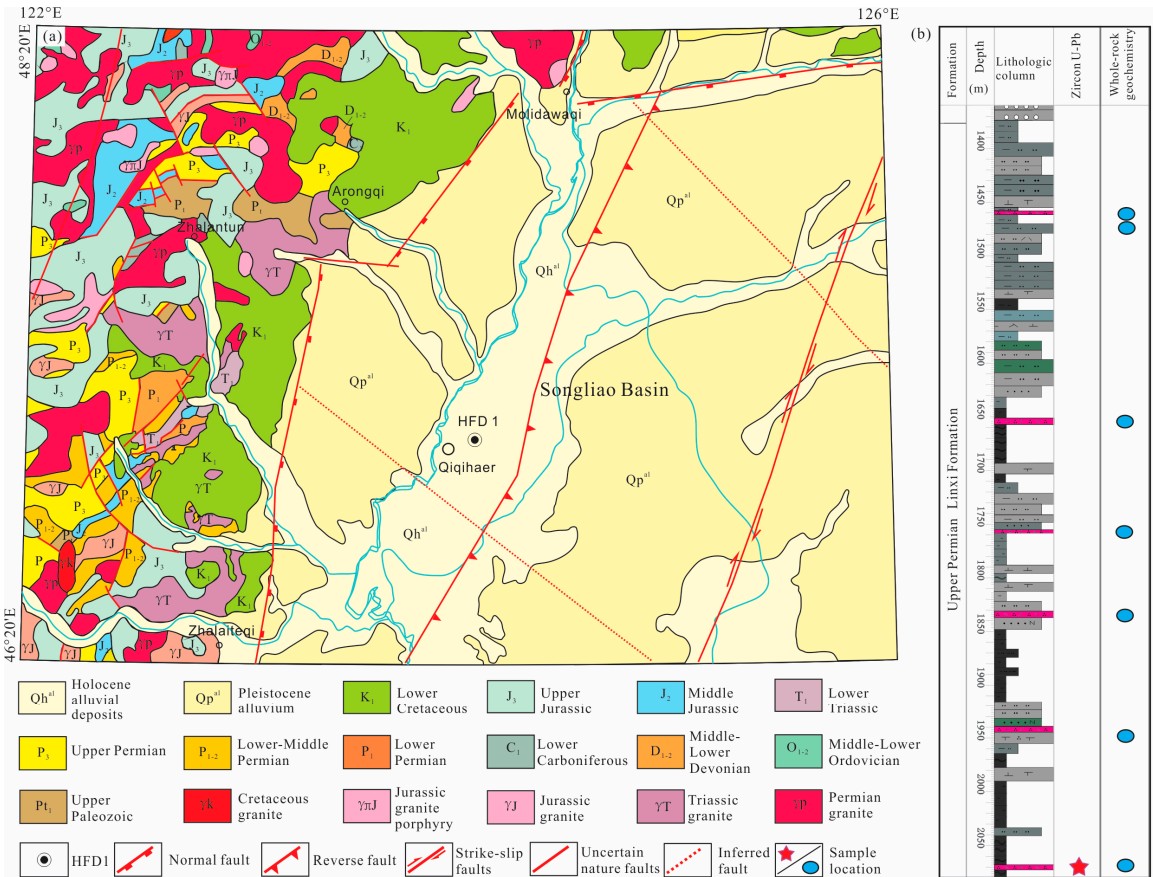

**Figure 2.** (**a**) Geological map of the Songliao Basin and adjacent region. (**b**) Borehole HFD1 comprehensive histogram (modified from [8]). The red star represents the geochronological sample, and the blue circles represent geochemical samples.

## 3. Sampling and Analytical Methods

### 3.1. Samples

One sample was used for isotopic analyses and seven for major and trace element analyses in this study. The lithology of both the dating samples and the rock geochemical analysis samples is granite. The rocks are light gray in color, exhibit subhedral to anhedral granular textures, and display blocky textures. The intrusive bodies form vein-like intrusions into the Permian Linxi Formation clastic rocks (Figure 2b).

Microscopic examination of the samples reveals that the rocks are primarily composed of plagioclase, K-feldspar, quartz, biotite, and opaque minerals. Plagioclase exhibits subhedral to euhedral tabular shapes, with visible polysynthetic twinning and albite perthitic twinning. It often displays zoning structures and undergoes slight sericitization and chloritization alteration, with more intense alteration towards the center. The grain size ranges from 0.20 to 2.00 mm, comprising approximately 45% of the sample. K-feldspar mostly exhibits poor euhedral development, appearing as anhedral grains with interstitial distributions. Some display euhedral to subhedral tabular shapes, with visible Carlsbad twinning and slight sericitization and chloritization. It forms micrographic textures with quartz, with particle sizes ranging from 0.06 to 1.25 mm, constituting approximately 30% of the sample. Quartz appears as anhedral grains, with clean surfaces and interstitial distributions. The grain size ranges from 0.03 to 0.30 mm, comprising approximately 15% of the sample. Biotite occurs as elongated or platy shapes, exhibiting brown to light brown pleochroism, parallel extinction, and third-order interference colors. It undergoes chloritization alteration, with particle sizes ranging from 0.06 to 0.25 mm, constituting approximately 5% of the sample. Opaque minerals are black in color, irregular in shape, mostly occurring as interstitial grains, with some appearing as anhedral grains, constituting less than 5% of the sample (Figure 3).

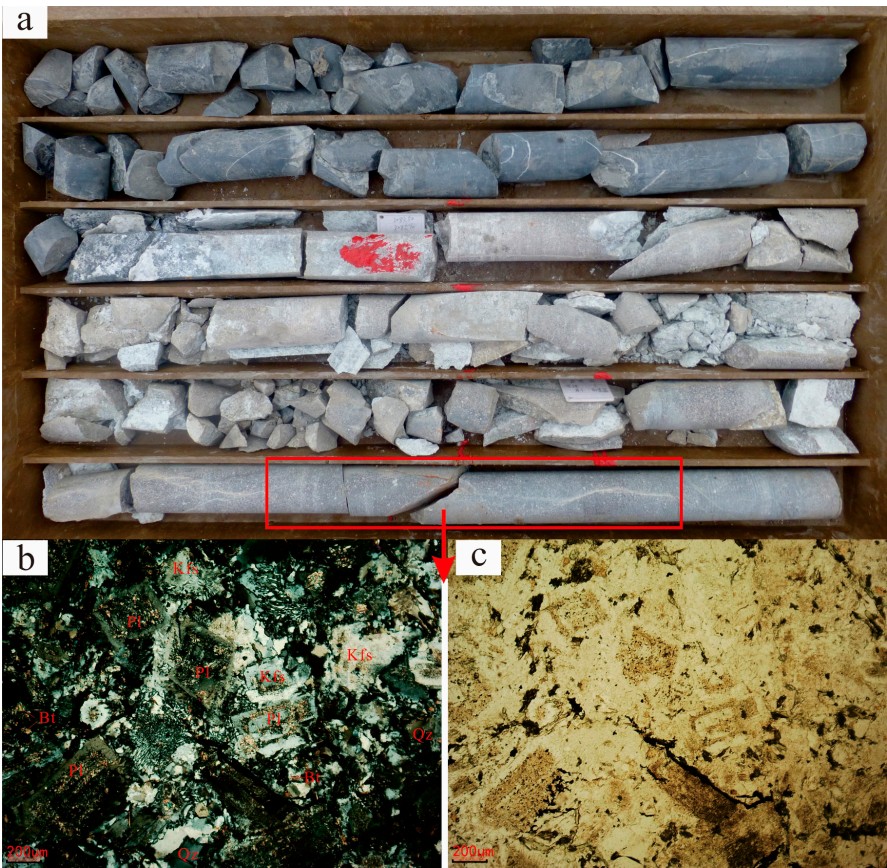

**Figure 3.** (**a**) Core photographs and (**b**,**c**) microscopic photographs of the granite (2088.3 TWS). Pl refers to Plagioclase, Kfs refers to Orthoclase, Qz refers to Quartz, Bt refers to Biotite.

### 3.2. Zircon Geochronology

Zircon grains were extracted from the samples by the Langfang Chengxin Geological Service Company, China, employing a combined magnetic and heavy liquid separation technique. Subsequently, these grains underwent examination under reflected and transmitted light using an optical microscope. Further analysis involved obtaining cathodoluminescence (CL) images utilizing a CL spectrometer (Garton Mono CL$^{3+}$) (AMETEK, Berwyn, PA, USA) installed on a Quanta 200F scanning electron microscope (SEM) (FEI company, Hillsboro, OR, USA) at Peking University. Operating parameters included an accelerating voltage of 15 kV, a beam current of 120 nA, and a scan time of 2 min. Based on the CL images, distinct domains within the grains were selected for U-Pb analysis.

The U–Pb geochronological analyses of the zircon grains were conducted using a quadrupole ICP–MS (Agilent 7500a) (Agilent Technologies, Santa Clara, CA, USA) equipped with a UP 193 solid-state laser (New Wave™ Research, Fremont, CA, USA) at the Geologic Laboratory Center, China University of Geosciences, Beijing. Laser parameters included a laser energy density of 8.5 J/cm$^2$, a repetition rate of 10 Hz, and a beam diameter of 36 μm. Ablation material was transported to the ICP–MS using high-purity He gas at a flux of 0.8 L/min. The 91,500 zircon served as the external standard, while the NIST 610 standard silicate glass was employed for instrument optimization. Additionally, the TEMORA zircon standard from Australia [39] and the Qinghu zircon standard from China [40] were utilized as secondary standards to monitor instrument drift.

Data analysis was performed using the GLITTER software (version 4.4, Macquarie University) to calculate isotopic ratios and element contents. Age calculations and concordia plots were generated using Isoplot 3.0 [41]. Common Pb correction was carried out following the Andersen method [42].

### 3.3. Whole-Rock Geochemistry

Major oxide compositions were determined using a PANalytical Axios Advance (PW4400) X-ray fluorescence (XRF) spectrometer (Malvern Panalytical, Worcestershire, United Kingdom) at the Northeast China Supervision and Inspection Center of Mineral Resources. Loss on ignition was assessed by heating 1 g of powder to 1100 °C for 1 h. The analysis of major oxides was conducted on fused glass with a precision of less than 2%.

Trace element compositions were analyzed using a PerkinElmer SCIEX ELAN 6000 ICP–MS (PerkinElmer, Massachusetts, the USA) at the same facility. For this analysis, 50 mg of each powdered sample was dissolved in a mixture of $HNO_3$ + HF in a high-pressure Teflon bomb for 48 h at 190 °C [43]. Rhodium (Rh) was utilized as an internal standard to calibrate instrumental drift during counting, while the international standard GBPG-1 was employed for quality control. The results of measurements of the GBPG-1 [44] and OU-6 [45] international standards were found to be in agreement with the recommended values. Overall, the precision of all elements was less than 5%.

## 4. Analytical Results

### 4.1. Zircon U-Pb Results

As observed in the CL image (Figure 4a), the zircons exhibit subhedral to euhedral shapes, with most retaining the original crystal morphology of magmatic crystalline zircons. They are predominantly elongated or short prismatic, with grain sizes mainly concentrated in the range 80–150 μm. The aspect ratio of length to width is mostly 2:1, and zircons display clear magmatic oscillatory zoning. Some zircons show varying degrees of fragmentation, with Th/U ratios ranging from 0.32 to 1.02 (Table 1). These characteristics indicate that the zircons in the samples are of magmatic origin.

**Table 1.** The U-Pb isotopic composition of the granite.

| Analytical Spot | Content/ppm | | | Th/U | Isotope Ratio | | | | | | Age/Ma | | | | | |
|---|---|---|---|---|---|---|---|---|---|---|---|---|---|---|---|---|
| | Pb | Th | U | | $^{207}Pb/^{206}Pb$ | ±1σ | $^{207}Pb/^{235}U$ | ±1σ | $^{206}Pb/^{238}U$ | ±1σ | $^{207}Pb/^{206}P$ | ±1σ | $^{207}Pb/^{235}U$ | ±1σ | $^{206}Pb/^{238}U$ | ±1σ |
| 1 | 3 | 57 | 56 | 1.02 | 0.05296 | 0.00529 | 0.30010 | 0.02941 | 0.04111 | 0.00099 | 327 | 177 | 266 | 23 | 260 | 6 |
| 2 | 3 | 45 | 56 | 0.81 | 0.07614 | 0.01438 | 0.43257 | 0.07918 | 0.04121 | 0.00207 | 1099 | 296 | 365 | 56 | 260 | 13 |
| 3 | 34 | 193 | 608 | 0.32 | 0.05348 | 0.00156 | 0.30194 | 0.00880 | 0.04095 | 0.00053 | 349 | 43 | 268 | 7 | 259 | 3 |
| 4 | 5 | 58 | 86 | 0.67 | 0.05873 | 0.00408 | 0.33273 | 0.02270 | 0.04110 | 0.00078 | 557 | 116 | 292 | 17 | 260 | 5 |
| 5 | 3 | 40 | 45 | 0.88 | 0.0627 | 0.00804 | 0.34340 | 0.04313 | 0.03972 | 0.00103 | 698 | 287 | 300 | 33 | 251 | 6 |
| 6 | 18 | 214 | 393 | 0.55 | 0.06322 | 0.00398 | 0.35574 | 0.02157 | 0.04081 | 0.00070 | 716 | 138 | 309 | 16 | 258 | 4 |
| 7 | 7 | 46 | 89 | 0.51 | 0.05771 | 0.00454 | 0.46520 | 0.03585 | 0.05847 | 0.00125 | 519 | 132 | 388 | 25 | 366 | 8 |
| 8 | 3 | 42 | 48 | 0.87 | 0.04663 | 0.00546 | 0.26499 | 0.03054 | 0.04122 | 0.00102 | 30 | 205 | 239 | 25 | 260 | 6 |
| 9 | 8 | 152 | 169 | 0.90 | 0.09699 | 0.02173 | 0.55122 | 0.11878 | 0.04122 | 0.00270 | 1567 | 319 | 446 | 78 | 260 | 17 |
| 10 | 28 | 133 | 353 | 0.38 | 0.0524 | 0.00741 | 0.29540 | 0.04084 | 0.04088 | 0.00136 | 303 | 250 | 263 | 32 | 258 | 8 |
| 11 | 2 | 30 | 41 | 0.72 | 0.05298 | 0.00774 | 0.30075 | 0.04299 | 0.04117 | 0.00141 | 328 | 259 | 267 | 34 | 260 | 9 |
| 12 | 43 | 514 | 758 | 0.68 | 0.05122 | 0.00202 | 0.28958 | 0.01131 | 0.04101 | 0.00058 | 251 | 64 | 258 | 9 | 259 | 4 |
| 13 | 4 | 36 | 72 | 0.50 | 0.05436 | 0.00359 | 0.30898 | 0.02004 | 0.04122 | 0.00074 | 386 | 114 | 273 | 16 | 260 | 5 |
| 14 | 3 | 49 | 53 | 0.93 | 0.05222 | 0.00612 | 0.29687 | 0.03416 | 0.04123 | 0.00108 | 295 | 211 | 264 | 27 | 260 | 7 |
| 15 | 2 | 33 | 48 | 0.68 | 0.05507 | 0.00724 | 0.31284 | 0.04032 | 0.04120 | 0.00123 | 415 | 235 | 276 | 31 | 260 | 8 |
| 16 | 3 | 38 | 49 | 0.77 | 0.05696 | 0.00478 | 0.32181 | 0.02628 | 0.04098 | 0.00081 | 490 | 192 | 283 | 20 | 259 | 5 |
| 17 | 3 | 29 | 61 | 0.48 | 0.06350 | 0.01102 | 0.35165 | 0.05983 | 0.04016 | 0.00137 | 725 | 389 | 306 | 45 | 254 | 8 |
| 18 | 2 | 19 | 30 | 0.62 | 0.06135 | 0.01047 | 0.34923 | 0.05824 | 0.04128 | 0.00167 | 652 | 296 | 304 | 44 | 261 | 10 |
| 19 | 4 | 59 | 84 | 0.70 | 0.04384 | 0.00495 | 0.24989 | 0.02778 | 0.04134 | 0.00103 | −79 | 173 | 226 | 23 | 261 | 6 |
| 20 | 3 | 42 | 49 | 0.85 | 0.05107 | 0.00331 | 0.28990 | 0.01849 | 0.04117 | 0.00072 | 244 | 114 | 258 | 15 | 260 | 4 |
| 21 | 7 | 106 | 138 | 0.77 | 0.04954 | 0.00341 | 0.28140 | 0.01905 | 0.04119 | 0.00076 | 173 | 119 | 252 | 15 | 260 | 5 |
| 22 | 3 | 32 | 43 | 0.75 | 0.07325 | 0.00507 | 0.40614 | 0.02711 | 0.04021 | 0.00073 | 1021 | 144 | 346 | 20 | 254 | 5 |
| 23 | 2 | 35 | 43 | 0.83 | 0.05240 | 0.00391 | 0.29743 | 0.02191 | 0.04117 | 0.00075 | 303 | 135 | 264 | 17 | 260 | 5 |
| 24 | 2 | 38 | 43 | 0.90 | 0.05352 | 0.00388 | 0.30525 | 0.02184 | 0.04136 | 0.00075 | 351 | 130 | 270 | 17 | 261 | 5 |

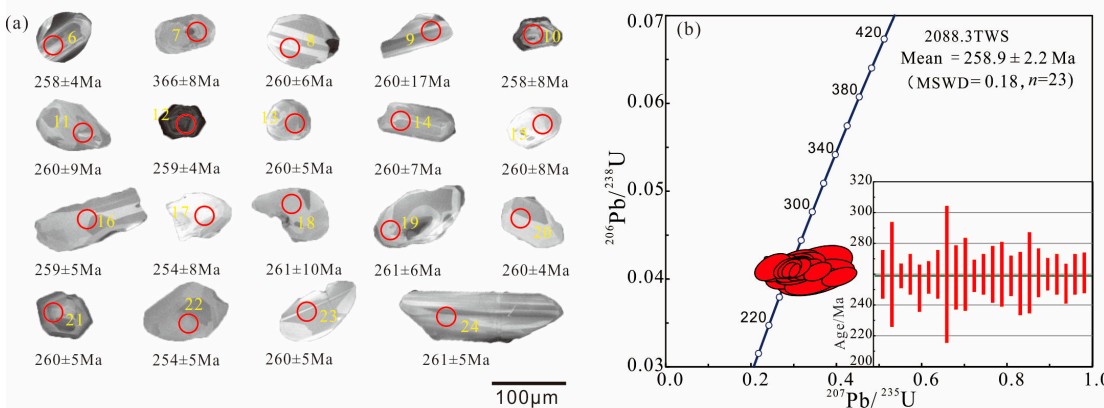

**Figure 4.** (**a**) CL images and $^{206}Pb/^{238}U$ ages of the granite. (**b**) Zircon U-Pb concordia diagram of the granite. Red circles indicate analytical spots for dating.

The analytical results from all 24 analytical spots fall on or near the concordia line, showing a relatively concentrated distribution (Figure 4b). Only one analytical point shows an older age (spot 7, 366 ± 8 Ma), likely representing inherited zircon ages. The $^{206}Pb/^{238}U$ weighted mean age of the remaining 23 analysis points is 258.9 ± 2.2 Ma (MSWD = 0.18, *n* = 23) (Figure 4b). Combining the above analyses, it can be concluded that the age of 258.9 ± 2.2 Ma represents the crystallization age of the magmatic rocks, indicating that the intrusion occurred during the early Late Permian.

### 4.2. Whole-Rock Geochemistry

Major Oxides

The results of major and trace element analyses for the granite samples are shown in Table 2. The $SiO_2$ content of the samples ranges from 70.38 wt.% to 72.23 wt.%, with an average of 71.11 wt.%, indicating acidic composition. On the TAS diagram (Figure 5a), the samples fall within the granite field. The $Al_2O_3$ content ranges from 15.38 wt.% to 15.71 wt.%, with an average of 15.54 wt.%. The MgO content varies from 0.77 wt.% to 1.09 wt.%, with an average of 0.92 wt.%. The CaO content ranges from 2.36 wt.% to 2.78 wt.%. The total alkali content ranges from 6.43 wt.% to 7.56 wt.%, with an average of 7.35 wt.%, indicating a distinct enrichment of $Na_2O$ over $K_2O$ and a $Na_2O/K_2O$ ratio ranging from 1.89 to 2.53. The rocks are classified within the calc-alkaline series (Figure 5b). The average CaO content is 2.54%. The aluminum saturation index A/CNK ranges from 0.97 to 1.14, with an average of 1.02, while the A/NK ratio ranges from 1.38 to 1.69, with an average of 1.46. These data indicate that the rocks are peraluminous and have relatively high alkali content.

The granite samples from the study area exhibit overall light rare-earth element (LREE) enrichment and heavy rare-earth element (HREE) depletion on the standardized distribution diagrams (Figure 6a,b). The content of rare-earth elements (ΣREE) ranges from 64.39 ppm to 69.28 ppm, with an average of 67.13 ppm. The total mass fraction of LREE (ΣLREE) ranges from 60.42 ppm to 65.16 ppm, with an average of 62.98 ppm. The total mass fraction of HREE (ΣHREE) ranges from 3.96 ppm to 4.29 ppm, with an average of 4.14 ppm. The LREE/HREE ratio ranges from 14.92 to 15.78, with an average of 15.20, indicating fractionation between LREE and HREE, with LREE enrichment. The standardized rare-earth element distribution curves exhibit similar right-skewed shapes, with $(La/Yb)_N$ ranging from 21.07 to 23.76, with an average of 21.29. The δCe values range from 0.90 to 0.94, with an average of 0.92, showing no significant anomalies. The δEu values range from 1.29 to 1.54, with an average of 1.40, indicating a noticeable positive anomaly, suggesting that there was no significant plagioclase crystallization separation during magma crystallization, distinguishing it from the upper continental crust characterized by significant negative Eu anomalies.

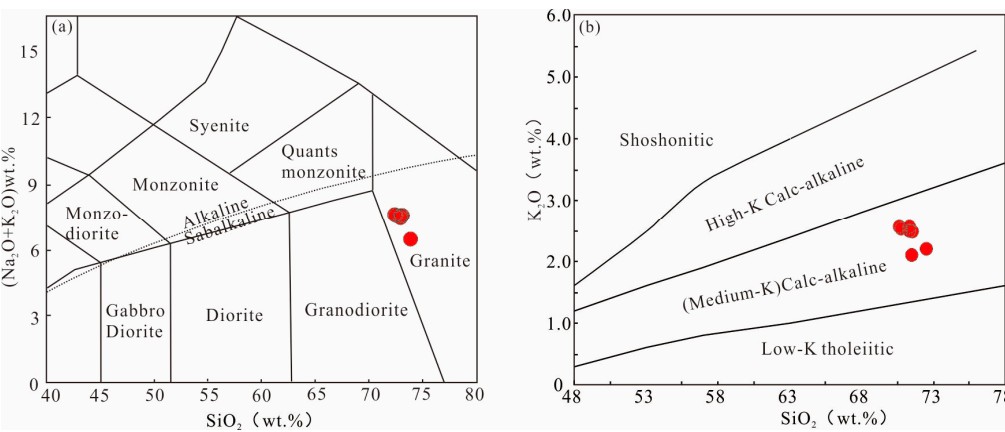

**Figure 5.** TAS (**a**) and $K_2O$-$SiO_2$ (**b**) diagrams of the granite.

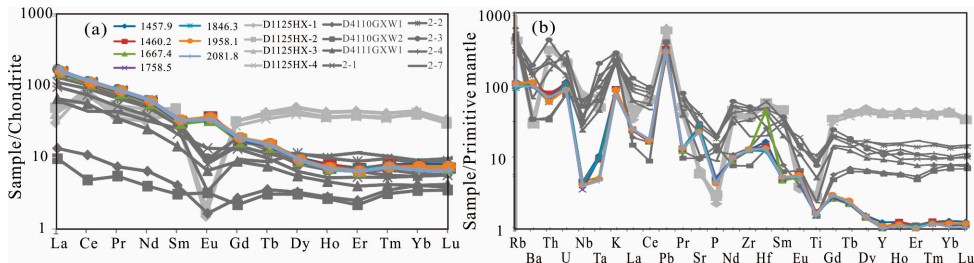

**Figure 6.** Chondrite-normalized REE patterns (**a**) and primitive mantle-normalized trace element spidergrams of granite (**b**). The samples represented by colored legends are from this study, while those represented by gray legends are from previously published samples [46,47].

On the normalized trace element diagrams (Figure 6a,b), the granite exhibits significant enrichment in large ion lithophile elements (LILEs) such as Rb, Pb, Sr, and Ba, while showing depletion in high field strength elements (HFSEs) including Nb, Ta, Ti, Zr, and Ce. The Sr mass fraction in the granite ranges from 448.29 ppm to 533.11 ppm, with an average of 499.68 ppm, exceeding 400 ppm. The Yb mass fraction ranges from 0.49 ppm to 0.59 ppm, with an average of 0.54 ppm, below 2 ppm, while Y ranges from 4.23 ppm to 5.19 ppm, with an average of 4.49 ppm. These characteristics classify the rock as high-Sr, low-Yb granite, indicative of formation under significant pressure [50,51]. On the Sr/Y-Y and $(La/Yb)_N$-$Yb_N$ diagrams (Figure 7a,b), all samples points fall within the adakitic rock field.

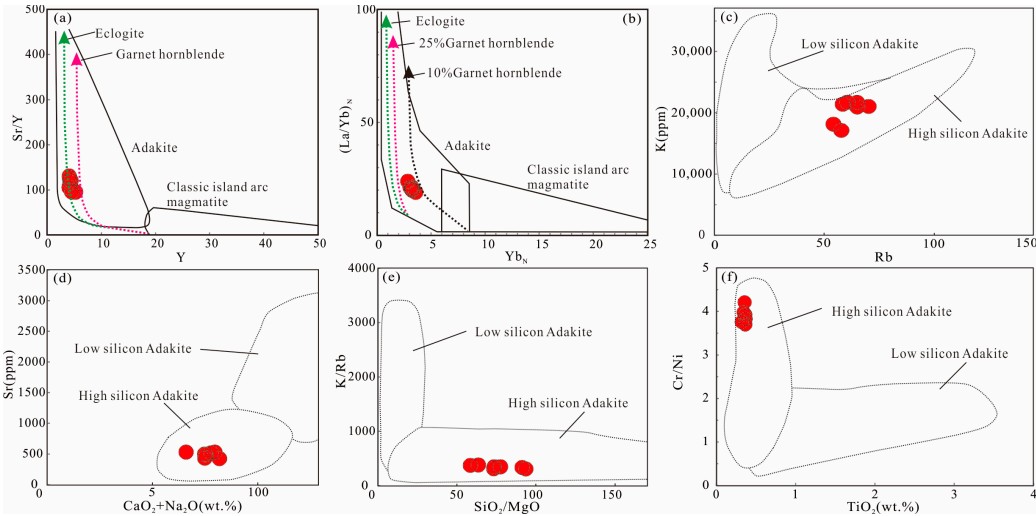

**Figure 7.** Sr/Y-Y (**a**) and $(La/Yb)_N$-$Yb_N$ (**b**) discrimination diagrams of adakites (base map after [52,53]). Discrimination diagrams of high silicon and low silicon adakites (**c**–**f**) (base map after [54]).

Table 2. Major and trace elemental compositions of magmatic rocks in northeastern China (wt.% for major oxides and ppm for trace elements).

| Sample | HFD1 | HFD2 | HFD3 | HFD4 | HFD5 | HFD6 | HFD7 | D4-1 * | D4-2 * | D4-3 * | 2-1 * | 2-2 * | 2-3 * | 2-4 * | 2-7 * | D1-1 * | D1-2 * | D1-3 * | D1-4 * |
|---|---|---|---|---|---|---|---|---|---|---|---|---|---|---|---|---|---|---|---|
| $SiO_2$ | 70.5 | 70.1 | 69.6 | 69.7 | 71.5 | 70.2 | 70.3 | 74.2 | 75.4 | 73.3 | 70.1 | 71.6 | 70.9 | 69.2 | 74.1 | 77.1 | 76.2 | 75.6 | 76.9 |
| $TiO_2$ | 0.33 | 0.34 | 0.34 | 0.34 | 0.30 | 0.32 | 0.31 | 0.04 | 0.04 | 0.20 | 0.37 | 0.50 | 0.33 | 0.41 | 0.07 | 0.07 | 0.08 | 0.09 | 0.08 |
| $Al_2O_3$ | 15.5 | 15.3 | 15.4 | 15.4 | 15.6 | 15.3 | 15.2 | 14.9 | 13.9 | 15.1 | 15.0 | 13.9 | 14.5 | 15.4 | 14.2 | 12.1 | 12.8 | 12.9 | 12.8 |
| $Fe_2O_3$-T | 2.00 | 1.96 | 2.05 | 2.16 | 1.78 | 2.08 | 2.08 | 2.01 | 2.06 | 2.98 | 2.98 | 2.88 | 2.21 | 2.69 | 1.25 | 1.41 | 0.69 | 1.16 | 0.83 |
| MnO | 0.04 | 0.04 | 0.04 | 0.04 | 0.04 | 0.04 | 0.04 | 0.08 | 0.04 | 0.04 | 0.06 | 0.05 | 0.05 | 0.05 | 0.07 | 0.04 | 0.02 | 0.03 | 0.03 |
| MgO | 0.93 | 0.92 | 1.08 | 1.02 | 0.90 | 0.78 | 0.76 | 0.22 | 0.25 | 0.43 | 0.80 | 0.63 | 0.54 | 0.96 | 0.21 | 0.05 | 0.09 | 0.14 | 0.08 |
| CaO | 2.48 | 2.46 | 2.75 | 2.62 | 2.45 | 2.33 | 2.52 | 0.16 | 0.32 | 0.26 | 2.21 | 1.25 | 1.06 | 2.50 | 0.29 | 0.16 | 0.23 | 0.29 | 0.25 |
| $Na_2O$ | 4.84 | 4.85 | 4.92 | 4.91 | 4.16 | 4.94 | 5.30 | 4.72 | 4.15 | 3.77 | 4.10 | 3.19 | 3.47 | 4.14 | 3.29 | 3.84 | 4.08 | 3.89 | 3.90 |
| $K_2O$ | 2.50 | 2.55 | 2.56 | 2.53 | 2.21 | 2.47 | 2.10 | 3.55 | 3.77 | 3.83 | 3.30 | 3.73 | 5.16 | 3.44 | 5.09 | 4.67 | 4.90 | 4.68 | 4.76 |
| $P_2O_5$ | 0.10 | 0.10 | 0.10 | 0.10 | 0.09 | 0.09 | 0.09 | 0.06 | 0.08 | 0.07 | 0.12 | 0.10 | 0.11 | 0.11 | 0.04 | 0.01 | 0.01 | 0.01 | 0.01 |
| LOI | 0.92 | 0.92 | 0.66 | 1.19 | 0.81 | 1.17 | 1.27 | 0.77 | 0.66 | 1.05 | 0.56 | 1.70 | 1.33 | 0.61 | 1.14 | 0.73 | 0.75 | 0.82 | 0.88 |
| Total | 100.2 | 99.6 | 99.6 | 100.0 | 99.8 | 99.7 | 100.0 | 100.8 | 100.8 | 101.2 | 99.6 | 99.7 | 99.7 | 99.6 | 99.8 | 100.3 | 99.9 | 99.6 | 100.6 |
| $K_2O/Na_2O$ | 1.94 | 1.90 | 1.92 | 1.94 | 1.89 | 2.00 | 2.53 | 1.33 | 1.10 | 0.98 | 1.24 | 0.86 | 0.67 | 1.20 | 0.65 | 0.82 | 0.83 | 0.83 | 0.82 |
| $SiO_2/Al_2O_3$ | 4.57 | 4.59 | 4.52 | 4.52 | 4.60 | 4.60 | 4.63 | 4.99 | 5.43 | 4.86 | 4.69 | 5.15 | 4.88 | 4.49 | 5.24 | 6.37 | 5.93 | 5.88 | 6.01 |
| $Fe_2O_3$ + MgO | 1.55 | 1.57 | 1.76 | 1.57 | 1.20 | 1.08 | 1.08 | 1.53 | 1.64 | 2.25 | 1.13 | 2.74 | 1.91 | 1.94 | 0.76 | 1.07 | 0.35 | 0.72 | 0.45 |
| Mg# | 0.39 | 0.40 | 0.43 | 0.38 | 0.38 | 0.30 | 0.30 | 0.21 | 0.24 | 0.24 | 0.23 | 0.39 | 0.36 | 0.35 | 0.22 | 0.09 | 0.17 | 0.18 | 0.14 |
| Sc | 4.07 | 3.52 | 3.51 | 3.61 | 4.19 | 3.82 | 3.50 | 1.55 | 1.20 | 1.62 | — | — | — | — | — | 4.15 | 3.70 | 4.66 | 4.22 |
| V | 39.5 | 38.5 | 38.3 | 35.1 | 39.3 | 32.1 | 35.2 | — | — | — | — | — | — | — | — | 4.83 | 5.85 | 7.63 | 5.97 |
| Cr | 19.3 | 15.0 | 15.0 | 16.4 | 15.1 | 14.6 | 21.1 | 11.0 | 10.9 | 17.8 | — | — | — | — | — | 4.48 | 4.02 | 3.86 | 3.63 |
| Ni | 4.62 | 3.82 | 4.00 | 4.25 | 3.98 | 3.67 | 4.26 | — | — | — | 1.44 | 4.88 | 4.80 | 1.44 | 5.53 | 2.24 | 1.94 | 3.02 | 2.48 |
| Cu | 16.1 | 10.4 | 17.9 | 17.6 | 15.1 | 15.1 | 14.1 | 3.60 | 3.59 | 5.59 | — | — | — | — | — | 3.37 | 2.83 | 4.02 | 3.91 |
| Zn | 39.0 | 42.7 | 51.0 | 50.6 | 51.3 | 49.0 | 48.3 | — | — | — | — | — | — | — | — | 84.1 | 49.3 | 80.5 | 47.9 |
| Ga | 22.7 | 22.1 | 22.5 | 22.8 | 22.6 | 21.0 | 21.9 | — | — | — | — | — | — | — | — | 26.7 | 29.2 | 28.2 | 29.0 |
| Rb | 69.3 | 64.9 | 61.2 | 59.5 | 55.9 | 65.0 | 58.9 | 310 | 226 | 224 | 130 | 118 | 191 | 111 | 261 | 188 | 215 | 215 | 213 |
| Sr | 509 | 498 | 533 | 523 | 532 | 454 | 448 | 41.8 | 44.2 | 129 | 268 | 166 | 192 | 289 | 50.5 | 25.3 | 20.8 | 23.3 | 21.1 |
| Y | 5.2 | 4.4 | 4.3 | 4.3 | 4.5 | 4.6 | 4.2 | 5.3 | 4.9 | 8.9 | 12.8 | 16.2 | 10.9 | 12.5 | 17.6 | 69.3 | 61.9 | 70.1 | 60.7 |
| Zr | 134 | 133 | 132 | 132 | 131 | 135 | 134 | 36.5 | 17.1 | 118 | 177 | 165 | 203 | 191 | 58.2 | 131 | 135 | 139 | 129 |
| Nb | 2.98 | 3 | 3.06 | 2.34 | 2.84 | 2.73 | 2.62 | 16.9 | 13.2 | 4.85 | 5.79 | 9.02 | 7.24 | 7.03 | 13.0 | 23.3 | 18.0 | 21.5 | 18.1 |
| Cs | 1.86 | 1.62 | 2.03 | 1.62 | 1.85 | 2.37 | 2.13 | 14.8 | 17.5 | 15.4 | — | — | — | — | — | 3.26 | 6.11 | 6.76 | 6.46 |
| Ba | 693 | 674 | 705 | 649 | 702 | 751 | 657 | 73.9 | 75.4 | 522 | 596 | 459 | 723 | 514 | 179 | 76.4 | 61.1 | 68.8 | 59.3 |
| Co | 5.34 | 5.72 | 6.44 | 5.98 | 6.98 | 5.67 | 5.52 | — | — | — | 4.20 | 4.86 | 3.10 | 4.04 | 2.27 | 2.64 | 2.43 | 2.41 | 2.51 |
| La | 16.2 | 15.5 | 15.5 | 15.5 | 16.3 | 16.0 | 16.1 | 3.45 | 2.50 | 16.4 | 23.7 | 27.4 | 41.2 | 32.2 | 15.5 | 7.37 | 12.1 | 9.90 | 9.87 |
| Ce | 29.3 | 28.0 | 27.7 | 27.8 | 29.5 | 28.8 | 30.2 | 7.26 | 3.31 | 36.4 | 47.6 | 58.1 | 73.3 | 64.9 | 28.7 | 53.1 | 43.9 | 56.4 | 47.4 |
| Pr | 3.44 | 3.33 | 3.18 | 3.32 | 3.48 | 3.47 | 3.46 | 0.78 | 0.58 | 3.54 | 5.20 | 6.67 | 8.86 | 7.33 | 4.38 | 3.51 | 5.27 | 4.39 | 3.97 |
| Nd | 12.1 | 11.8 | 11.3 | 11.8 | 12.5 | 12.3 | 12.4 | 3.33 | 2.08 | 12.5 | 19.2 | 24.8 | 30.8 | 25.8 | 16.2 | 15.3 | 22.6 | 19.0 | 17.5 |
| Sm | 2.16 | 2.01 | 1.96 | 2.05 | 2.11 | 2.18 | 2.15 | 0.7 | 0.53 | 2.38 | 3.49 | 4.66 | 5.29 | 4.69 | 4.19 | 6.52 | 7.63 | 6.96 | 5.85 |

**Table 2.** *Cont.*

| Sample | HFD1 | HFD2 | HFD3 | HFD4 | HFD5 | HFD6 | HFD7 | D4-1 * | D4-2 * | D4-3 * | 2-1 * | 2-2 * | 2-3 * | 2-4 * | 2-7 * | D1-1 * | D1-2 * | D1-3 * | D1-4 * |
|---|---|---|---|---|---|---|---|---|---|---|---|---|---|---|---|---|---|---|---|
| Eu | 0.81 | 0.92 | 0.81 | 0.91 | 0.86 | 0.89 | 0.85 | 0.11 | 0.21 | 0.42 | 0.85 | 0.52 | 0.61 | 0.57 | 0.22 | 0.09 | 0.1 | 0.12 | 0.1 |
| Gd | 1.55 | 1.50 | 1.46 | 1.50 | 1.60 | 1.66 | 1.64 | 0.63 | 0.51 | 1.99 | 2.95 | 3.20 | 4.15 | 3.31 | 2.13 | 6.64 | 6.72 | 6.78 | 5.81 |
| Tb | 0.24 | 0.23 | 0.22 | 0.23 | 0.24 | 0.25 | 0.24 | 0.15 | 0.13 | 0.31 | 0.40 | 0.47 | 0.52 | 0.46 | 0.36 | 1.64 | 1.50 | 1.62 | 1.32 |
| Dy | 1.05 | 0.99 | 0.97 | 0.98 | 1.01 | 1.01 | 1.00 | 0.94 | 0.88 | 1.55 | 1.80 | 3.18 | 2.27 | 2.59 | 3.08 | 13.3 | 11.5 | 12.9 | 10.4 |
| Ho | 0.19 | 0.19 | 0.17 | 0.17 | 0.17 | 0.18 | 0.17 | 0.18 | 0.17 | 0.29 | 0.33 | 0.61 | 0.42 | 0.5 | 0.65 | 2.52 | 2.14 | 2.46 | 2.07 |
| Er | 0.50 | 0.49 | 0.45 | 0.48 | 0.50 | 0.46 | 0.45 | 0.47 | 0.41 | 0.74 | 0.98 | 1.61 | 1.18 | 1.18 | 2.11 | 7.58 | 6.54 | 7.44 | 6.34 |
| Tm | 0.09 | 0.09 | 0.08 | 0.08 | 0.08 | 0.08 | 0.08 | 0.10 | 0.09 | 0.12 | 0.15 | 0.27 | 0.16 | 0.19 | 0.29 | 1.09 | 0.95 | 1.08 | 0.91 |
| Yb | 0.59 | 0.55 | 0.54 | 0.52 | 0.54 | 0.54 | 0.49 | 0.78 | 0.66 | 0.78 | 1.04 | 1.56 | 1.03 | 1.13 | 1.71 | 7.85 | 6.92 | 7.63 | 7.44 |
| Lu | 0.09 | 0.08 | 0.08 | 0.08 | 0.08 | 0.08 | 0.07 | 0.11 | 0.10 | 0.12 | 0.16 | 0.25 | 0.17 | 0.19 | 0.27 | 0.88 | 0.78 | 0.85 | 0.82 |
| Hf | 4.82 | 4.19 | 13.20 | 4.15 | 3.56 | 3.88 | 3.77 | 2.24 | 0.97 | 4.24 | 4.98 | 6.94 | 5.06 | 2.35 | 1.67 | 5.98 | 7.11 | 6.57 | 6.67 |
| Ta | 0.41 | 0.36 | 0.20 | 0.38 | 0.37 | 0.19 | 0.18 | 1.70 | 0.96 | 0.64 | 0.98 | 1.26 | 4.57 | 2.77 | 2.26 | 1.26 | 0.75 | 1.22 | 1.00 |
| Pb | 21.4 | 22.1 | 21.6 | 20.0 | 20.7 | 20.1 | 21.8 | 10.8 | 19.5 | 19.9 | — | — | — | — | — | 40.6 | 37.2 | 42.3 | 37.5 |
| Th | 5.70 | 6.29 | 5.16 | 5.24 | 4.73 | 4.83 | 5.57 | 0.99 | 0.89 | 1.89 | 9.40 | 12.1 | 25.5 | 9.90 | 7.00 | 19.6 | 19.1 | 19.4 | 20.8 |
| U | 2.26 | 2.10 | 2.10 | 2.11 | 2.11 | 1.76 | 1.81 | 0.63 | 0.62 | 1.10 | 2.90 | 1.40 | 2.30 | 1.50 | 3.90 | 2.63 | 2.59 | 3.25 | 2.80 |
| ΣREE | 68.3 | 65.7 | 64.4 | 65.5 | 68.9 | 67.9 | 69.3 | 19.0 | 12.2 | 77.5 | 108 | 133 | 170 | 145 | 79.7 | 127 | 129 | 138 | 120 |
| LREE | 64.0 | 61.5 | 60.4 | 61.5 | 64.6 | 63.6 | 65.2 | 15.6 | 9.21 | 71.6 | 100 | 122 | 160 | 135 | 69.1 | 85.8 | 91.6 | 96.8 | 84.6 |
| HREE | 4.29 | 4.12 | 3.96 | 4.04 | 4.22 | 4.25 | 4.13 | 3.36 | 2.95 | 5.90 | 7.81 | 11.15 | 9.90 | 9.55 | 10.6 | 41.5 | 37.1 | 40.8 | 35.1 |
| LREE/HREE | 14.9 | 15.0 | 15.3 | 15.2 | 15.3 | 15.0 | 15.8 | 4.65 | 3.12 | 12.1 | 12.8 | 11.0 | 16.2 | 14.2 | 6.52 | 2.07 | 2.47 | 2.37 | 2.41 |
| $La_N/Yb_N$ | 19.8 | 20.4 | 20.7 | 21.6 | 21.7 | 21.1 | 23.8 | 3.17 | 2.72 | 15.1 | 16.4 | 12.6 | 28.7 | 20.4 | 6.48 | 0.67 | 1.25 | 0.93 | 0.95 |
| δEu | 1.29 | 1.54 | 1.41 | 1.51 | 1.37 | 1.37 | 1.32 | 0.50 | 1.22 | 0.57 | 0.79 | 0.39 | 0.38 | 0.42 | 0.20 | 0.04 | 0.04 | 0.05 | 0.05 |
| δCe | 0.91 | 0.91 | 0.92 | 0.90 | 0.92 | 0.90 | 0.94 | 1.04 | 0.65 | 1.12 | 1.01 | 1.02 | 0.90 | 1.00 | 0.84 | 2.55 | 1.35 | 2.09 | 1.86 |

Note: δCe = Ce/Ce * = Ce $_{CN}$/(La $_{CN}$ × Pr $_{CN}$) $^{1/2}$; δEu = Eu/Eu * = Eu $_{CN}$/(Sm $_{CN}$ × Gd $_{CN}$) $^{1/2}$. N = Chondrite normalized; the normalization value after [48,49]. LOI: loss ion ignition. The data marked with "*" are from [46,47].

## 5. Discussion

### 5.1. Crystallization Age

In this study, U-Pb zircon dating analysis was conducted on granite samples from well HFD1 in the northern Songliao Basin. The sampled rocks are preserved and unmetamorphosed, with zircons displaying clear magmatic oscillatory zoning. Additionally, the geochemical characteristics of the rocks confirm their magmatic origin. All 23 data points obtained from the samples fall on the U-Pb concordia line, representing the age of magmatic intrusion and crystallization (Figure 4b). Therefore, the age of the granite in the study area is determined to be $258.9 \pm 2.2$ Ma (MSWD = 0.18, $n$ = 23), indicating an early Late Permian intrusive event.

### 5.2. Rock Type and Petrogenesis

5.2.1. Identification of Adakite

The adakite rock type was initially identified by Defant and Drummond [52] as volcanic or plutonic rocks derived from partial melting of young (<25 Ma) hot oceanic crust under eclogite facies conditions. Later, it broadly referred to intermediate-acidic rocks with specific geochemical characteristics, characterized by Sr > 3 ppm, Y < 20 ppm, Yb < 2 ppm, and MgO < 3 wt.% [50,51,55]. Subsequently, scholars proposed dividing adakites into O-type and C-type based on their geochemical characteristics. O-type adakites, associated with slab partial melting, require higher pressures for formation and are characterized by Na enrichment, representing the classical adakites. On the other hand, C-type adakites, enriched in K, possibly originate from partial melting of thickened continental crust (>50 km) or delaminated lower crust mafic rocks, unrelated to slab melting [56–63]. Some researchers suggest that adakites may result from complex mantle-derived mafic magma high-pressure crystallization in island arc settings [64–66]. Based on geochemical characteristics, Martin et al. [53] classified adakites into High-$SiO_2$ adakites and Low-$SiO_2$ adakites. High-$SiO_2$ adakites mainly originate from subducted oceanic crust, while Low-$SiO_2$ adakites are derived from mantle peridotite and typically exhibit higher MgO contents.

Considering the above studies, the formation of adakites can be summarized into three main types: (1) partial melting of subducted oceanic slab; (2) basaltic intrusion and partial melting of mantle or lower crust ultramafic and mafic rocks; (3) crystallization under high-pressure conditions in the initial arc magma of island arc settings [67,68]. The classification of adakites indicates that O-type adakites and High-$SiO_2$ adakites share similar chemical characteristics, both representing Na-enriched oceanic crust-derived adakitic rocks, consistent with the original definition of adakites, referred to as "typical adakites" [54].

The granite in the northern part of the Songliao Basin has high $SiO_2$ and $Al_2O_3$ contents, characteristic of Na-rich and K-poor features. The high $Na_2O/K_2O$ ratio suggests a possible origin from low-K tholeiitic primary magmas, often formed in island arc or active continental margin environments, mainly related to the melting of oceanic crust. The rock has high Sr content but low Yb and Y, displaying typical geochemical characteristics of adakites. Additionally, the samples exhibit high Sr/Y ratios (98–125, average 112) and La/Yb ratios (average 21.3), consistent with typical features of adakite. Thus, the granite studied here is considered adakitic rock (Figure 7a,b).

From the harker diagrams, it can be observed that the adakite samples in this study were mostly located in the area of subducted oceanic crust melting and thickened lower crust melting (Figure 8). However, with relative Na enrichment and K depletion, an average $SiO_2$ content of 71.11 wt.%, low MgO content, low Th content (4.73–6.29 ppm), and low Th/Ce ratios, the samples exhibit LREE enrichment, HREE depletion, and lack of negative Eu anomalies, indicating a distinct difference from C-type adakites formed by partial melting of mafic rocks in the lower crust. C-type adakites are characterized by significant K enrichment. In the Cr-Ni diagram, the samples display low Cr and Ni contents, reflecting characteristics of slab melting (Figure 9a), while also exhibiting characteristics of O-type adakite.

The intrusive rocks studied in this research exhibit highly consistent patterns in their rare-earth elements and trace elements, showing pronounced light–heavy rare-earth fractionation, which distinguishes them from island arc calc-alkaline series rocks [68]. Analysis using La-(La/Yb)$_N$ and Sm-La/Sm diagrams similarly reveals significant partial melting features in the studied adakitic rocks, without significant crystal fractionation, indicating they are not products of primary basaltic magma separation (Figure 9b,c). Empirical diagrams, including K-Rb, Sr-(CaO + Na$_2$O), K/Rb-SiO$_2$/MgO, and Cr/Ni-TiO$_2$, for high-silica and low-silica adakitic rock classification (Figure 7c–f), consistently categorize the rock samples as high-silica adakitic rocks. Based on these analyses, we conclude that the granite studied here represents typical adakitic rock, exhibiting characteristics of O-type adakitic and SiO$_2$-rich adakite rocks.

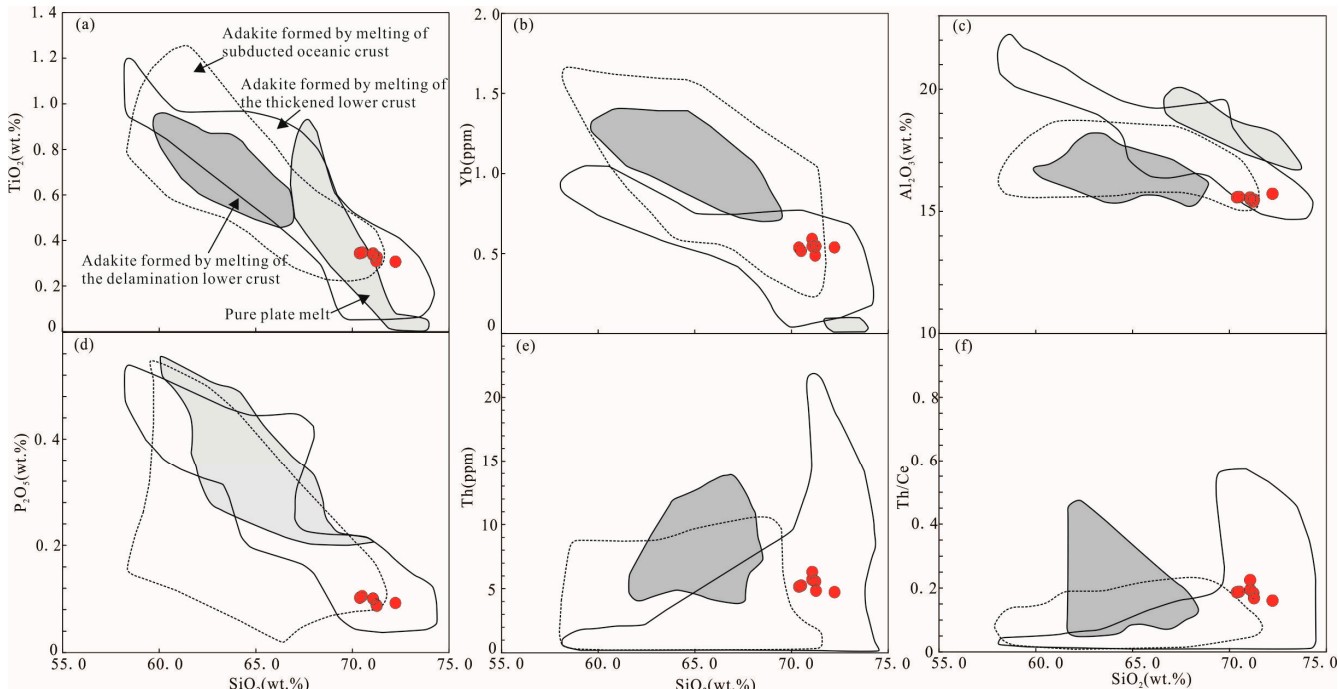

**Figure 8.** Harker diagrams for adakites from the study area (base map from [69]). (**a**) SiO$_2$ versus TiO$_2$, (**b**) SiO$_2$ versus Yb, (**c**) SiO$_2$ versus Al$_2$O$_3$, (**d**) SiO$_2$ versus P$_2$O$_5$, (**e**) SiO$_2$ versus Th, and (**f**) SiO$_2$ versus Th/Ce.

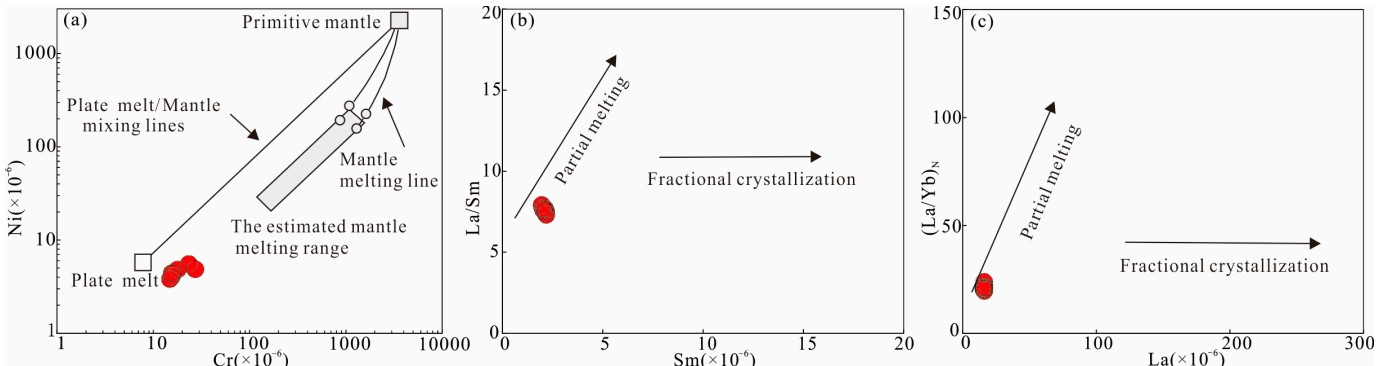

**Figure 9.** (**a**) Ni–Cr diagrams for quartz monzonit (base map from [70]), (**b**) La/Sm–Sm and (**c**) (La/Yb)N–La diagrams for adakites (base map after [62]).

### 5.2.2. Petrogenesis

The O-type adakitic rocks, enriched in Na, are primarily associated with subduction processes of tectonic plates, partly related to the partial melting of low-K mantle plume

basalts or thickened lower crust with oceanic crust characteristics [71]. From the original mantle trace element spider diagram (Figure 6b), the granite is enriched in large ion lithophile elements and depleted in high field strength elements, with relative enrichments in Th, Sr, Zr, and U, resembling features of adakitic rocks formed by subducted oceanic crust melting [72–75]. The ΣREE of the samples is relatively low, showing significant fractionation with enrichment in LREE and depletion in HREE, along with parallel distribution curves, reflecting characteristics of magmatic evolution from a single source. The presence of a significant positive Eu anomaly indicates minimal or absent strong or multistage crystallization differentiation processes, consistent with features reflected in Sm-La/Sm and La-(La/Yb)N diagrams (Figure 9b,c), further confirming the granite as an O-type adakitic rock.

According to Zhang et al. [71], Na-enriched adakitic rocks have three main genetic types: partial melting of subducted plate MORB, composition of oceanic crust, and lower crustal intrusion of low-K basalt. As previously described, the geochemical attributes of the rocks suggest that the Na enrichment is attributed to partial melting of subducted plate MORB, indicating that the granite was generated from the melting of subducted plate MORB during the late stage of oceanic subduction. Based on adakitic rock discrimination diagrams such as $SiO_2$–Yb and $SiO_2$–$TiO_2$ (Figure 8), the samples mostly fall within the adakitic rock zone formed by subducted oceanic crust melting, further supporting the model of subducted oceanic crustal plate melting.

Experimental studies indicate that the formation of O-type adakitic rocks requires high pressure. Rapp et al. [76] reported the results of basaltic melting experiments at different pressure stages ranging from 0.8 to 3.2 GPa. With increasing pressure, the residual phases also change systematically. Peacock et al. [77] suggested that adakitic rock formation occurs within a narrow pressure range (1.8–2.3 GPa, equivalent to depths of 60–70 km), with residual phases consisting of garnet + clinopyroxene + amphibole. Therefore, O-type adakitic rocks are typically interpreted as being formed by partial melting of subducted oceanic crust at depths of 75–85 km (corresponding to the garnet amphibolite–eclogite transition zone) [52,77,78].

In the Y-Sr/Y diagram (Figure 7a), all adakitic rock sample points are located between the garnet amphibolite and garnet amphibolite–garnet clinopyroxenite evolution lines, indicating that the source minerals of the adakitic rocks (granite) in this study are related to garnet and clinopyroxene. This reveals that the original rocks of the studied adakitic rocks should be garnet amphibolite or amphibolite metamorphic rock series [79], and Zhang and Jiao. [63] also pointed out that the solid solution in equilibrium with garnet and eclogite is a fundamental attribute of adakitic rocks. The granite has Y/Yb ratios of 7.94–8.79 and (Ho/Yb)N ratios of 0.97–1.05, with right-skewed HREE curves, indicating the presence of roughly equal proportions of garnet and clinopyroxene in the residual phase, with depleted Y, Yb, and HREE, suggesting the stable presence of garnet during partial melting and unstable presence of clinopyroxene in the source region, and almost no or little plagioclase in the residual minerals, indicating magma crystallization [52,71]. Therefore, it is inferred that the Late Permian adakitic rocks originated from oceanic crustal plates undergoing garnet amphibolite metamorphism during late Paleozoic subduction to the lower crust. As the oceanic crustal plate subducted deeper into the subduction zone, partial melting occurred in the garnet amphibolite metamorphic zone, generating adakitic magmas, which then ascended and intruded to form the adakitic rocks.

### 5.3. Tectonic Implications

Adakitic rocks have been regarded as a special rock type with significant tectonic implications since their proposal [52], and were believed to be products of plate melting in a subduction context. The genesis of adakitic rocks represents the tectonic environment in which they formed, providing unequivocal evidence for crustal subduction and plate convergence processes. The geochemical characteristics and genesis of the granite in this study indicate that the rock was formed by partial melting of MORB-type subducted

oceanic crust. Combined with previous research [8,9,11,80], it is generally believed that the Paleo-Asian Ocean closed from the Late Permian to the Early-Middle Triassic, indicating that the adakitic rocks formed during the Early Late Permian were derived from melting of subducted slabs at depth, implying that in the Early Late Permian, the Paleo-Asian Ocean in the study area had not closed and was likely in the late stage of subduction.

Combined with data from adjacent areas, this study provides a comprehensive constraint on the tectonic evolution of the Paleo-Asian Ocean. The Paleo-Asian Ocean has experienced the process from subduction, collision to post-closure crustal extension in the study area. Each process has different rock records. The author studied Triassic granites developed in the Horqin Righ Middle Banner and Ulanhot areas of Inner Mongolia (Figure 10a; [46,47]), and conducted a comprehensive discussion on the evolutionary process of the Paleo-Asian Ocean. The age of the granites in Horqin Right Middle Banner is 242.6 Ma, indicating their emplacement during the Early-Middle Triassic. The geochemical characteristics suggest they are S-type granites (Figure 6a,b), indicating their formation in a regionally compressed tectonic background and thickened crust caused by collisional orogenic activity, recording the collisional amalgamation time of the Paleo-Asian Ocean in this area (Figure 10b) [47]. Reports on Triassic magmatic events in the Linxi area in the southern part of the study area indicate the development of intrusions at Linxi and Shuangjing, with ages of 237.2 ± 2.7 Ma and 229.2 ± 4.1 Ma, respectively. These granites are likely derived from thickened crust in orogenic belts and re-melting of relatively old continental margins, suggesting the beginning of the collisional amalgamation of the Paleo-Asian Ocean in the mid-Permian and its end in the mid-Triassic [11].

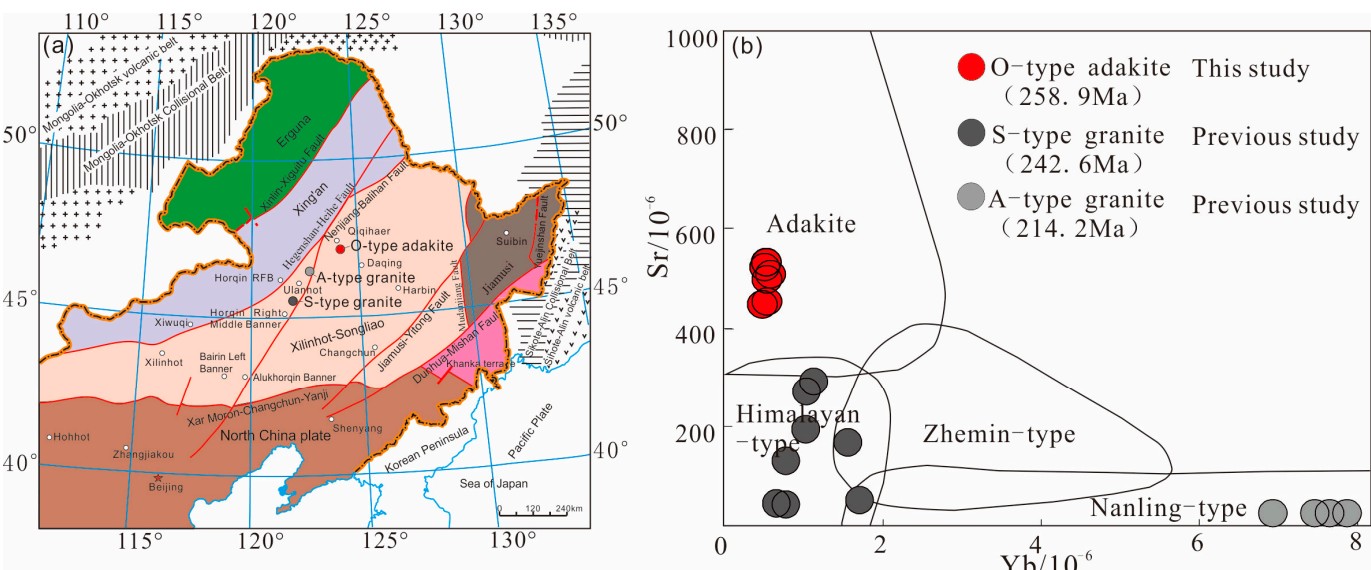

**Figure 10.** (**a**) The distribution and (**b**) Sr−Yb classification diagram of the granite. (**a**) Base map from Zhang et al. [9]. The reference data from Zhang et al. [46,47].

In the Ulanhot region, there are occurrences of biotite monzogranite with a crystallization age of 214.4 Ma and an emplacement age in the middle-late Triassic [46]. These rocks exhibit strong negative Eu anomalies and significant Sr depletion, characteristic of low-Sr, high-Yb granite (Figure 6a,b). This granite is interpreted as an A2-type post-orogenic granite formed in a post-orogenic extensional tectonic setting (Figure 10b), indicating the conclusion of collision during the Middle Triassic and the onset of post-orogenic extension in the late Triassic [46,47]. Ge et al. [81] reported Triassic granites in the Ulanhot region (235–225 Ma), associating them with lithospheric extension after the closure of the Paleo-Asian Ocean. Li et al. [11] proposed that magmatism in the northern margin of the North China Block and adjacent areas occurred in two stages during the Early-Mid and Late Triassic, with late Triassic intrusive rocks around 220 Ma reflecting regional extensional tectonics related to

late-stage orogenic evolution, signifying the termination of orogenesis and the beginning of a new tectonic evolution stage.

Combining the regional research findings with our results and previous publications [46,47], we infer that the O-type adakite (granite) formed in the early Late Permian (approximately 258.9 Ma) represents the final stage of subduction of the Paleo-Asian Oceanic slab, formed by partial melting of the subducted Paleo-Asian Ocean crustal slab at specific depths (under high-pressure conditions). This indicates that the Paleo-Asian Ocean had not closed by the early Late Permian and represents the terminal stage of subduction (Figure 11). The S-type granites in the middle Triassic (242.6 Ma) in the nearby Korqin Youyi Zhongqi area suggest the closure of the Paleo-Asian Ocean and thickening of the crust by the early to middle Triassic. The emplacement of the late Triassic (214.4 Ma) biotite monzogranite in the Ulanhot area indicates that the region had completed its evolution before the late Triassic, transitioning to a post-orogenic extensional stage, marking the onset of a new tectonic domain—the Circum-Pacific tectonic domain—and recording the evolution from collision to extension after the North China–Siberian collision.

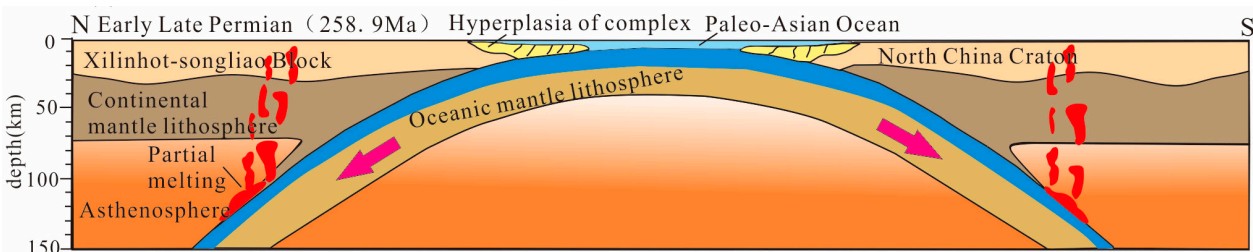

**Figure 11.** Tectonic evolution model of the study area during Late Permian.

## 6. Conclusions

1. The U-Pb age of adakite is 258.9 ± 2.2 Ma, representing the timing of emplacement and crystallization, indicating the magmatism in the early Late Permian.
2. The adakite is characteristic of typical O-type adakite. The adakite is enriched in light rare-earth elements with a relatively consistent parallel flat pattern of heavy rare-earth elements, reflecting characteristics of magmatic evolution from a single source and slab melting.
3. We propose that the adakite is derived from the melting of subducted MORB-type ocean crust during the subduction of the Paleo-Asian Oceanic slab. This indicates that in the early Late Permian in northeastern China, the Paleo-Asian Ocean had not closed, and oceanic subduction continued until 258.9 ± 2.2 Ma. In the early Late Permian (258.9 Ma), the Paleo-Asian Ocean was still in the subduction phase, forming typical O-type adakite under high-pressure conditions.

**Author Contributions:** Conceptualization, H.Z., L.Q. and J.Z.; Data curation, H.Z.; Investigation, H.Z., J.Z., S.C. and Y.Z. (Yuejuan Zheng); Methodology, H.Z. and L.Q.; Software, L.Q. and J.Z.; Validation, H.Z. and Y.M.; Writing—original draft, H.Z., L.Q. and J.Z.; Writing—review and editing, H.Z., L.Q., Y.M., H.D. and Y.Z. (Yujin Zhang). All authors have read and agreed to the published version of the manuscript.

**Funding:** This study was supported by funding from the China Geological Survey Project (Grant DD20240034, DD20230220 and DD20230210). National Natural Science Foundation of China (NSFC 42172024, 42372269).

**Data Availability Statement:** Data are contained within the article.

**Acknowledgments:** We thank Wenchun Ge for assistance in isotope analysis. We appreciate the constructive comments from two anonymous reviewers. We thank Xiangyu Dong, Xiaolong Wu and Wenxiu Li for their help in the drilling process.

**Conflicts of Interest:** The authors declare no conflicts of interest.

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
