# Peer review of "Discovery of Late Permian Adakite in Eastern Central Asian Orogenic Belt: Implications for Tectonic Evolution of Paleo-Asian Ocean"

_minerals, doi:10.3390/min14040386_

Round 1

Reviewer 1 Report

Comments and Suggestions for Authors

In this paper, the authors have discovered late Permian adakite in the Songliao Basin, which is also a critical area for understanding the evolution of the Paleo-Asian Ocean. The data presented are credible and important. However, the authors obtained only one zircon age from the Late Permian, and the conclusion is also based on this data. Why are there three diagrams in the tectonic settings in Figures 11? I question the validity of Figures 11b and 11c, which are based on previous research. If that's the case, you should include them in detail or provide a brief discussion. In Figure 10, previous data are suddenly added into the paper without clear introduction in the preceding sections. Therefore, the discussion in section 5.3 on Tectonic implications should be rewritten. I suggest the authors focus on the Permian data or provide a detailed introduction of previous data. Consequently, a major revision should be completed for the paper.

Specific comments:

Line 20: delete “was”

Line 26: change the “Adakite” to “adakite”

Line 26: revise “was formed by” into “formed through”

Line 51: Solonker-Silinhe-Changchun-Yanji? Silinhe? Xar Moron?

Line 60: Please move the first sentence to Line 91 and combine them. You introduce the location of the study area twice.

Line 86: Replace “experiencing the overlay and influenc of” with “which was influenced by the overlay of”

Line 95: delete one blank space.

Line 97: Linsixi or Linxi? Check the other parts

Line 98-102: Modify the depth range from large values to small values. Also, ensure consistency in the placement of decimal points.

Line110-111: exchange the “structures” with “textures”

Line112: Linsixi or Linxi?

Line132: photographsof should be photographs of

Line144: delete “the” before “China University of Geosciences”

Line 171: “cathodoluminescence” should be “CL”

Line 175: they? What is the subject of the sentence?

Line 185: The numbers should be the superscripts in "206Pb/238U".

Line 193: delete “rare earth”

Line 207: add “is” before “from”

Line 226: lack “is”

Line 231: What is the source of the data? Clarify them clearly.

Line 258-302: change the “Adakite” to “adakite” in the entire manuscript. Why do you always capitalize the first letter of “Adakite(s)”?

Line 259: The following sentence lacks a subject.

Line 293: change Harke to harker

Line 324: The font should be consistent with the others.

Line 376: You should focus on the Permian data in the following part. The previous data have already been introduced in prior research. Why are they being discussed together here again? Rewrite the discussion in section 5.3.

Line 400-419: replace “ancient Asian Ocean” with “Paleo-Asian Ocean”.

Line 466: rewrite the conclusions. They are too long and redundant.

Figures and tables:

Figure 1: add Hegenshan-Heihe fault.

Figure 3: TWS? Please give its full name first. Please ensure that the full names of minerals are included in the figure captions, as abbreviations are used in the diagrams.

Figure 5: It is not correct that you use diagrams of intrusive and volcanic rocks together. The rock names should be revised in Fig 5b.

Figure 10: Why add S-type and A-type granites here suddenly?

Figure 11: You have only got one zircon age. Why are there three diagrams in the tectonic settings? I question the validity of Figures 11b and 11c, which are based on previous research. 

Table 1: ensure consistency in the placement of decimal points.

Table 2: Please ensure that the decimal points are consistent with the precision suggested in your paper, especially for trace elements. You should replace "133.79" with "133", replace "21.43" with "21.4", "12.14" with "12.1", and so on.

Comments on the Quality of English Language

The Quality of English Language is good.

Author Response

Dear Reviewer:

 Many thanks to the reviewer for the constructive comments and suggestions. We have carefully revised the manuscript following each reviewer's comments and suggestions, and have responded to each of them. We have deleted Figures 11b and 11c. Thanks for your suggestions and comments.

In order to comprehensively define and clearly discuss the evolutionary history of the Paleo-Asian Ocean, the article combines previous research results and adds explanations and discussions in the article, thus defining the evolution process of the ancient Asian Ocean from subduction to collision, and then to post-orogenic extension.

We have made major changes to Section 5.3 and revision for the paper. Please see the modification text for details. Thank you so much.

Specific comments:

Line 20: delete “was”

Reply: Thanks. We have deleted it. Please see new line 25 in the revised manuscript.

Line 26: change the “Adakite” to “adakite”

Reply: We have corrected it. Please see new line 32.Thanks.

Line 26: revise “was formed by” into “formed through”

Reply: We have corrected it. Please see new line 33. Thanks.

Line 51: Solonker-Silinhe-Changchun-Yanji? Silinhe? Xar Moron?

Reply: Thanks. We have corrected it. It is “Solonker-Xar Moron-Changchun-Yanji”. Please see new line 61.

Line 60: Please move the first sentence to Line 91 and combine them. You introduce the location of the study area twice.

Reply: We have corrected it. Please see new lines 105-109.Thanks.

Line 86: Replace “experiencing the overlay and influenc of” with “which was influenced by the overlay of”

Reply: We have corrected it. Please see new lines 100-101.Thanks.

Line 95: delete one blank space.

Reply: We have deleted it. Please see new line 115. Thanks.

Line 97: Linsixi or Linxi? Check the other parts

Reply: Thanks. We have corrected it. It is “Linxi”. Please see new line 117. We have revised the entire text.

Line 98-102: Modify the depth range from large values to small values. Also, ensure consistency in the placement of decimal points.

Reply: Thanks. We have corrected it. Please see new lines 117-127.

Line110-111: exchange the “structures” with “textures”

Reply: We have corrected it. Please see new line 138. Thanks.

Line112: Linsixi or Linxi?

Reply: We have corrected it. Please see new line 139.Thanks.

Line132: “photographsof” should be “photographs of”

Reply: We have corrected it. Please see new line 160.Thanks.

Line144: delete “the” before “China University of Geosciences”

Reply: We have deleted it. Please see new line 194. Thanks.

Line 171: “cathodoluminescence” should be “CL”

Reply: We have corrected it. Please see new lines 235. Thanks.

Line 175: they? What is the subject of the sentence?

Reply: We have corrected it. Please see new line 239. Thanks.

Line 185: The numbers should be the superscripts in "206Pb/238U".

Reply: We have corrected it. Please see new line 250.Thanks.

Line 193: delete “rare earth”

Reply: We have deleted it. Please see new line 258.Thanks.

Line 207: add “is” before “from”

Reply: We have added it. Please see new lines 273-274.Thanks.

Line 226: lack “is”

Reply: We have corrected it. Please see new lines 292-294.Thanks.

Line 231: What is the source of the data? Clarify them clearly.

Reply: We have corrected it. Please see new line 304. Thanks.

Line 258-302: change the “Adakite” to “adakite” in the entire manuscript. Why do you always capitalize the first letter of “Adakite(s)”?

Reply: Thank you so much. We have corrected it. Please see new lines 332-378.

Line 259: The following sentence lacks a subject.

Reply: We have corrected it. Please see new line 333.Thanks.

Line 293: change Harke to harker

Reply: We have changed it. Please see new line 369. Thanks.

Line 324: The font should be consistent with the others.

Reply: We have corrected it. Please see new lines 400-401. Thanks.

Line 376: You should focus on the Permian data in the following part. The previous data have already been introduced in prior research. Why are they being discussed together here again? Rewrite the discussion in section 5.3.

Reply: Thanks. We have corrected it. We have made major changes to Section 5.3. Please see new Section 5.3. Thank you so much.

Line 400-419: replace “ancient Asian Ocean” with “Paleo-Asian Ocean”.

Reply: Thanks. We have changed it. Please see new lines 476-494.

Line 466: rewrite the conclusions. They are too long and redundant.

Reply: Thanks. We have corrected it. Please see new lines 551-567.

Figures and tables:

Figure 1: add Hegenshan-Heihe fault.

Reply: Thanks. We have added it. Please see new Figure 1.

Figure 3: TWS? Please give its full name first. Please ensure that the full names of minerals are included in the figure captions, as abbreviations are used in the diagrams.

Reply: Thank so much. TWS is the name of the sample. It has no specific meaning and is not the name of the mineral.

Figure 5: It is not correct that you use diagrams of intrusive and volcanic rocks together. The rock names should be revised in Fig 5b.

Reply: Thanks. We have corrected it. Please see new Figure 5.

Figure 10: Why add S-type and A-type granites here suddenly?

Reply: Thanks. In order to comprehensively limit and explore the evolutionary history of the ancient Asian Ocean, the article combines previous research results, thus defining the evolutionary process of the ancient Asian Ocean from subduction(O-type adakite) to collision(S-type granite) and to post-orogenic extension(A-type granite). We have already explained the background information, and have provided explanation and discussion in the article. We also changed the sample referenced in new Figure 10 to a lighter tone. Thank you so much.

Figure 11: You have only got one zircon age. Why are there three diagrams in the tectonic settings? I question the validity of Figures 11b and 11c, which are based on previous research. 

Reply: Thanks for the suggestion. We have deleted Figures 11b and 11c. Please see new Figure 11.Thank you so much.

Table 1: ensure consistency in the placement of decimal points.

Reply: Thanks. We have corrected it. Please see new Table 1.

Table 2: Please ensure that the decimal points are consistent with the precision suggested in your paper, especially for trace elements. You should replace "133.79" with "133", replace "21.43" with "21.4", "12.14" with "12.1", and so on.

Reply: Thanks. We have corrected it. Please see new Table 2.

Reviewer 2 Report

Comments and Suggestions for Authors

I went through the manuscript entitled "Discovery of late Permian adakite in Eastern Central Asian Orogenic Belt: Implications for tectonic evolution of Paleo-Asian Ocean" by Zhang and co-authors. The manuscript describes the discovery of a granite with typical SiO2-rich adakite signature in the Asian Orogenic belt. Geochemical data support considering such rocks as O-type adakites and the relevant trace element diagrams are nicely presented and convincing. Actually, the figures and diagrams can be refined since the font size is too small and details are difficult to appreciate. As regards the U-Pb dating results I have suggested to present the results using probability Kernel hystograms following the references given in the annotated pdf. The only part of the manuscript that is weak in the present form is that regarding the petrographic description of the samples, with mineral microanalysis that is completely lacking. From my point of view this must be addressed in the revised version. Finally, I think there is a great equivocation on the name of the studied rocks: if they contain > 70 wt% of SiO2, fall in the granite field in the TAS and also the authors at a certain point in the manuscript call them "granites", why these rocks are attributed as "quartz-monzonite" throughout the text? I have suggested to firstly define the rock as granite and to specify later that the geochemical fingerprints allow to characterize this granite as SiO2-rich adakite. I think this is more adequate. You will find my comments and suggestions in the annotated pdf attached.

Regards,

Reviewer

Comments on the Quality of English Language

The English is generally fine, though it can be improved and sentences can be shortened, going directly to the point.

Author Response

Dear Reviewer:

Thanks for the very helpful comments and suggestions, and the attached corrections annotated on our manuscript. We have carefully revised the manuscript following each reviewer's comments and suggestions, and have responded to each of them.

The figures and tables have been modified according to your opinions; regarding the U-Pb dating results, we carefully read and studied the reference you recommended. The lithology in the reference is clastic rock, while ours is for magmatic rocks. The possible age results of the presentation method will be slightly different; regarding the petrographic description of the sample, we do lack the electronic probe test. Unfortunately, I am currently a visting scholar at the University of Melbourne. Now the sampling and analysis testing work cannot be completed in the near future. We are very grateful to the reviewer for the constructive comment. We will continue to pay attention and work hard to do these analyses; regarding the name of the rock, we have modified it according to your opinions. Thank you so much.

We have modified all the issues you marked in the PDF file one by one. Please see the modification text for details. Thank you very much.

Specific comments:

Line 17: change the “and yet represents a scientific problem” to “ as an essential scientific issue”.

Reply: We have corrected it. Please see new line 22 in the revised manuscript. Thanks.

Line 18: change the “on” to “ for”.

Reply: We have corrected it. Please see new line 24.Thanks.

Line 20:Replace “shows that the quartz monzonite was formed” with “indicate crystallization of quartz monzonite”.

Reply: We have corrected it. Please see new lines 25-26.Thanks.

Line 21: Replace “Petro-logical and geochemical analysis show that t” with “T”.

Reply: We have corrected it. Please see new line 27.Thanks.

Line 25: Replace “typical” with “can be classified as”.

Reply: We have corrected it. Please see new line 31.Thanks.

Line 26: change the “Adakite” to “adakite”

Reply: We have corrected it. Please see new line 33.Thanks.

Line 27: Replace “that it was formed” with “derivation”.

Reply: We have corrected it. Please see new line 33.Thanks.

Line 27: change the “the” to “partial”

Reply: We have corrected it. Please see new line 33.Thanks.

Line 27: change the “plate” to “crust”

Reply: We have corrected it. Please see new line 34.Thanks.

Line 41: change the “Researchers suggest that t” to “T”

Reply: We have corrected it. Please see new line 49.Thanks.

Line 82-84:Replace “providing constraints and detailed, reliable data for understanding 82Late Paleozoic magmatic events, tectonic background, and the evolution of the Paleo-83Asian Ocean in the Songliao Basin” with “to understand the connection between the magmatic activity and the evolution of the Paleo-Asian Ocean in the Songliao Basin”.

Reply: We have corrected it. Please see new lines 96-97.Thanks.

Line 93: It is not clear to me what is intended with strata here. Are these strata a Carboniferous-Permian sedimentary cover?

Reply: We have corrected it. Please see new line 112.Thanks.

Line 94: Maybe it is better: "felsic magma emplacement"

Reply: We have corrected it. Please see new line 113.Thanks.

Line 95-96: Replace “into the Permian strata and conducted analysis and research on them” with “, which represent the subject of this study”.

Reply: We have corrected it. Please see new line 116.Thanks.

Line 97-102:Please provide details on rocks and sediments collected by the well; not the name of formations. International readers not confident with the area cannot understand what is meant here.

Reply: We have corrected it. Please see new lines 117-127.Thanks.

Line 108-110:Bad english writing. Do you mean that one sample was used for isotopic analyses and seven for major and trace element analyses?

Reply: We have corrected it. Please see new lines 134-135.Thanks.

Line 112:The size of the figure is too small: it is difficult to see anything.

Reply: We have corrected it. Please see new Figure 2b.Thanks.

Line 114: Replace “potassium” with “K-”.

Reply: We have corrected it. Please see new line 141.Thanks.

Line 116:Normal or reverse zoning? Anyway, microanalysis of plagioclase, K-feldspar, biotite and opaques in the sample are needed.

Reply: Thanks. We really appreciate the constructive suggestion. We do lack the electronic probe test. I am currently a visting scholar at the University of Melbourne, so the sampling and analysis testing work cannot be completed in the near future. We will continue to pay attention and work hard to do these analyses.

Line 118: Replace “particle” with “grain”.

Reply: We have corrected it. Please see new line 145.Thanks.

Line 119: Replace “potassium” with “K-”.

Reply: We have corrected it. Please see new line 146.Thanks.

Line 124: Replace “particle” with “grain”.

Reply: We have corrected it. Please see new line 152.Thanks.

Line 125-130:In general there is a few documentation (micrographs) of what is described here and microanalysis of minerals is mandatory.

Reply: We are very grateful to the reviewer for the constructive comment. We will continue to pay attention and work hard to do the microanalysis of minerals.

Line 152-153: Please note, histograms of Kernel density estimate of U-Pb spot ages as in Fornelli et al. (2020, 2022) should be preferred.

Reply:Thank you so much. We carefully read and studied the reference you recommended. The lithology in the reference is clastic rock, while ours is for magmatic rocks. The possible age results of the presentation method will be slightly different.

Line 187: Replace “formation” with “crystallization”.

Reply: We have corrected it. Please see new line 253.Thanks.

Line 196: It is not clear to me why you call these rocks quartz-monzonite when it is a granite. Maybe you can call these rocks also as SiO2-rich adakites given the trace element concentrations fitting with the adakite suite.

Reply: We have corrected it. Please see the modification text for details.Thanks.

Line 239-241: Therefore, I think you will agree with my previous comment and the correct name (granite) must be used throughout the text replacing "quartz-monzonite"

Reply: We have corrected it. Please see new modification text. Thanks.

Line 250: Replace “fresh” with “preserved”. 

Reply: We have corrected it. Please see new line 323.Thanks.

Line 362-363: This sentence must be better explicated. It is not clear what is intended with "the solid solution in equilibrium with garnet"

Reply: We have corrected it. Please see new line 438.Thanks.

Line 377-390: This part can be eliminated because it is repetition or it can be drastically shortened. Please go to the point, the time for reading papers has drastically shortened for academics...

Reply: We have corrected it. Please see new section 5.3.Thanks.

Line 392: Replace “plate MORB” with “crust”. 

Reply: We have corrected it. Please see new line 468.Thanks.

Line 446: Replace “own study” with “results”. 

Reply: We have corrected it. Please see new line 529.Thanks.

Line 477: Replace “MORB” with “MORB-type oceanic crust”. 

Reply: We have corrected it. Please see new line 563.Thanks.

Line 478: Replace “has” with “had”. 

Reply: We have corrected it. Please see new line 565.Thanks.

Round 2

Reviewer 1 Report

Comments and Suggestions for Authors

The data are crucial, and the discussions on the adakites found in the Songliao Basin are significant for understanding the evolution of the PAO. The authors have made substantial improvements to their manuscript. This vesion is easy to read, and certain contexts are now smoother to understand. Thus, I recommend accepting it after minor revisions.

Minor revisions:

Line 16:add “the” before “Northeast China”

Line: 19-20, “Zircon U-Pb dating indicate crystallization of granite at 258.9 ± 2.2 Ma, as the product of magmatism in the early Late Permian.” can be revised into” Zircon U-Pb dating indicates that granite crystallized at 258.9 ± 2.2 Ma, as the product of magmatism occured in the early Late Permian.

Line 23:Replace “ratio” into “ratios”.

Line: 57-58: volcanic magma events or volcanic-magma events are redundant, as "volcanic" already implies an association with magma and volcanic activity. Please revise them into magma events.

Line 313:grant? granite

Line 431:add “the” before “Northeast China”

Author Response

Dear Reviewer:

Thanks for the very helpful comments and suggestions. We have carefully revised the manuscript following each reviewer's comments and suggestions, and have responded to each of them. Please see the modification text for details. Thank you very much.

Minor revisions:

Line 16:add “the” before “Northeast China”

Reply: We have corrected it. Please see line 16.Thanks.

Line: 19-20, “Zircon U-Pb dating indicate crystallization of granite at 258.9 ± 2.2 Ma, as the product of magmatism in the early Late Permian.” can be revised into “Zircon U-Pb dating indicates that granite crystallized at 258.9 ± 2.2 Ma, as the product of magmatism occured in the early Late Permian.”

Reply: We have corrected it. Please see lines 19-20.Thanks.

Line 23:Replace “ratio” into “ratios”.

Reply: We have corrected it. Please see line 23.Thanks.

Line: 57-58: volcanic magma events or volcanic-magma events are redundant, as "volcanic" already implies an association with magma and volcanic activity. Please revise them into magma events.

Reply: We have corrected it. Please see new line 57.Thanks.

Line 313:grant? Granite

Reply: It is granite, and we have corrected it. Please see new line 314.Thanks.

Line 431:add “the” before “Northeast China”

Reply: We have corrected it. Please see new line 432.Thanks.